# Characterization of the Anti-Viral and Vaccine-Specific CD8^+^ T Cell Composition upon Treatment with the Cancer Vaccine VSV-GP

**DOI:** 10.3390/vaccines12080867

**Published:** 2024-08-01

**Authors:** Tamara Hofer, Lisa Pipperger, Sarah Danklmaier, Krishna Das, Guido Wollmann

**Affiliations:** 1Institute of Virology, Medical University of Innsbruck, A-6020 Innsbruck, Austria; tamara.hofer@i-med.ac.at (T.H.); lisa.pipperger@i-med.ac.at (L.P.); sarah.danklmaier@i-med.ac.at (S.D.); krishna.das@boehringer-ingelheim.com (K.D.); 2Christian Doppler Laboratory for Viral Immunotherapy of Cancer, A-6020 Innsbruck, Austria; 3Department of Internal Medicine V, Haematology & Oncology, Medical University Innsbruck, A-6020 Innsbruck, Austria; 4Tyrolean Cancer Research Institute, A-6020 Innsbruck, Austria; 5ViraTherapeutics GmbH, A-6063 Rum, Austria

**Keywords:** VSV-GP, cancer vaccine vector, CD8^+^ T cell effector phenotype, short-lived effector cells (SLECs), memory-precursor effector cells (MPECs), vaccine regime design, route of vaccination, vaccination dose, heterologous cancer vaccination

## Abstract

Numerous factors influence the magnitude and effector phenotype of vaccine-induced CD8^+^ T cells, thereby potentially impacting treatment efficacy. Here, we investigate the effect of vaccination dose, route of immunization, presence of a target antigen-expressing tumor, and heterologous prime-boost with peptide vaccine partner following vaccination with antigen-armed VSV-GP. Our results indicate that a higher vaccine dose increases antigen-specific CD8^+^ T cell proportions while altering the phenotype. The intravenous route induces the highest proportion of antigen-specific CD8^+^ T cells together with the lowest anti-viral response followed by the intraperitoneal, intramuscular, and subcutaneous routes. Moreover, the presence of a B16-OVA tumor serves as pre-prime, thereby increasing OVA-specific CD8^+^ T cells upon vaccination and thus altering the ratio of anti-tumor versus anti-viral CD8^+^ T cells. Interestingly, tumor-specific CD8^+^ T cells exhibit a different phenotype compared to bystander anti-viral CD8^+^ T cells. Finally, the heterologous combination of peptide and viral vaccine elicits the highest proportion of antigen-specific CD8^+^ T cells in the tumor and tumor-draining lymph nodes. In summary, we provide a basic immune characterization of various factors that affect anti-viral and vaccine target-specific CD8^+^ T cell proportions and phenotypes, thereby enhancing our vaccinology knowledge for future vaccine regimen designs.

## 1. Introduction

Upon viral infections and cancer therapy, cytotoxic CD8^+^ T cells (CTL) represent the main executors of target cell destruction by direct and indirect mechanisms. These include cell lysis through perforin and granzyme secretion, activation of pro-apoptotic cascades via Fas and TRAIL receptors on T-cells, and upregulation of MHC class I molecules on the surface of tumor cells via IFNγ release [1]. CD8^+^ T cells are activated through TCR and co-receptor stimulation caused by the interaction with antigen-loaded conventional dendritic cells type 1 (cDC1) directly leading to massive proliferation and clonal CD8^+^ T cell expansion [2]. CD8^+^ T cells require three signals for functional effector differentiation. In the first two steps, TCR signaling, together with co-receptor signaling via CD8, CD28, CD27, etc., stimulates CD8^+^ T cell activation. In addition, inflammatory cytokines like IFN-α and IL-12 provide a third signal that unleashes the full cytolytic potential of effector CD8^+^ T cells [3,4]. Upon primary prophylactic and therapeutic vaccination, approximately seven days are required to reach the peak of clonal CD8^+^ T cell expansion. Booster vaccinations lead to a significantly reduced time for secondary expansion to peak but also result in a diminished and delayed contraction phase, resulting in a longer period of peak levels [5,6]. Additionally, boost immunizations commonly induce higher quantities of antigen-specific CD8^+^ T cells, thereby increasing the chance for therapeutic efficacy [7]. However, high CD8^+^ T cell proportions do not always translate into treatment efficacy, and it has emerged that the quality of antigen-specific CD8^+^ T cells also impacts their therapeutic capacity [8].

One qualitative characteristic that can be measured upon CD8^+^ T cell induction is their ability to either become short-lived effector cells or differentiate into long-lived memory cells. The surface expression markers killer-cell lectin-like receptor G1 (KLRG-1) and interleukin receptor 7 (IL-7R/CD127) enable the discrimination of four different subtypes, namely short-lived effector cells (SLECs), memory-precursor effector cells (MPECs), early effector cells (EECs), and double-positive cells (DPs), a process initiated already early after CD8^+^ T cell activation [9,10]. KLRG-1 expressing cells are highly differentiated and will ultimately experience replicative senescence, while IL-7R expression activates Jak/STAT signaling, thereby supplying crucial survival signals for memory T cells [11,12]. Regarding their roles in infections, their designation as either SLECs or MPECs already indicates their main purpose, with the first delivering effector functions upon clonal expansion and the latter raising a pool of memory cells for long-term protection. In contrast, the roles of EECs and DPs were initially less clear when the classification of CD8^+^ T cell effector subtypes first arose, but have since been elucidated. EECs dominate the pool of antigen-specific CD8^+^ T cells several days before the peak of clonal expansion and are equally capable of inducing SLECs and MPECs [13]. On the contrary, DPs can lose KLRG-1 and have the potential to become part of the memory pool later after a pathogen encounter [14]. However, their relative contribution to the MPEC pool might be modest due to a decreased overall number compared to the prominent SLEC and MPEC pools. Intriguingly, many vaccine studies have focused on SLEC and MPEC phenotypes and have shown statistically significant differences between these two subtypes [15,16].

The pathogen-induced inflammatory environment is the main factor determining either SLEC or MPEC development [17]. Here, strong type I interferon (IFN α/ß) or IL-12 signaling induces inflammation-dependent expression of T-bet. A high expression of this transcription factor leads to the induction of KLRG-1 and thus SLEC development, while low expression promotes MPEC formation [9,17]. In addition, intranasal compared to intravenous inoculation with the vaccine vector vesicular stomatitis virus (VSV) showed that the respiratory environment enhanced the clonal expansion of anti-viral CD8^+^ T cells while memory CD8^+^ T cells were decreased. However, little is known regarding whether other routes of immunization, different doses, or the presence of a target-antigen expressing tumor entity influences CD8^+^ T cell frequencies and effector subtype compositions following viral vaccine vector applications.

Moreover, partly opposing results have been published indicating that either SLEC or MPEC proportions correlate with treatment efficacy in therapeutic cancer vaccine settings [16,18]. Whether there is one distinct subtype responsible for increased therapeutic success remains an open question in the field and might differ between model systems and malignancies.

VSV-GP is a genetically engineered VSV in which the glycoprotein has been replaced by the glycoprotein of the lymphocytic choriomeningitis virus (LCMV), thereby eliminating neurotoxicity and diminishing the induction of neutralizing antibodies in mice [19]. VSV-GP displays oncolytic potential in tumors with aberrant type-I IFN responses and is currently being evaluated in clinical phase I trials (NCT05155332) [20]. Moreover, VSV-based vectors and virus variants can be additionally modified to incorporate therapeutic transgenes making it an optimal vaccine vector [21,22]. For example, VSV-GP has been tested preclinically as a vaccine vector against various infectious pathogens, including human immunodeficiency virus (HIV) and respiratory syncytial virus (RSV) [23,24]. Furthermore, the heterologous combination of target antigen-expressing VSV-GP with the peptide vaccine KISIMA showed pre-clinical efficacy by strongly enhancing tumor-specific CD8^+^ T proportions and reshaping tumor immune infiltrating cell compositions in immunologically hot and cold tumors, and is currently being assessed in clinical phase I trials (NCT04046445, NCT05846516) [25,26]. Consequently, VSV-GP provides a promising and broadly applicable candidate for clinical applications, and therefore further research is desired.

Hence, we here investigate application-relevant factors of VSV-GP, including the virus’ dose, route of immunization, presence of an antigen-positive tumor entity, and combination with a heterologous vaccine partner in regard to CD8^+^ T cell quantity and quality. In the future, this basic immune activation characterization profile may aid in optimizing cancer vaccine treatment regimens.

## 2. Materials and Methods

### 2.1. Ethics Approval

Ethical approval for all animal experiments was obtained from the Institutional Animal Care and Use Committee (ZVTA) of the Medical University of Innsbruck and the Austrian Federal Ministry of Science, Research and Economy (BMWFW-66.011/0041-WF/V/3b/2016, BMBWF-66.011/0125-V/3b/2019, 2020-0.475.503). Studies were performed according to institutional guidelines of the Medical University of Innsbruck, Austria, and the national animal experimentation law. Mice were maintained in a BSL2 facility in individually ventilated cages at a temperature between 20 and 24 °C, with a light/dark cycle of 12 h, ad libitum feeding, and water.

### 2.2. Tumor Cell Line and Tumor Cell Implantation

B16F10-OVA cells kindly provided by K.Erb (Boehringer Ingelheim GmbH, Biberach, Germany) were cultured in complete DMEM (Sigma-Aldrich #D5671, Taufkirchen, Germany) medium containing 10% Fetal Bovine Serum (FBS; PAN-Biotech #P30-3306, Aidenbach, Germany), 100 U/mL penicillin and 100 µg/mL streptomycin (Gibco #15140-122, Carlsbad, CA, USA), 2% Glutamine (Gibco, #25030-024) and 3 µg/mL Puromycin (Gibco, #AT1138-03). For tumor cell implantation, 5 × 10^5^ cells per 100 µL PBS (Sigma-Aldrich #D8537) were injected subcutaneously (s.c.) into the right flank of six- to eight-week-old female C57BL/6JRj mice (Janvier Labs, Le Genest St Isle, France) under isoflurane anesthesia. Mice were monitored every two to three days and tumor volume was calculated with the following formula: 0.4 × length × width^2^. The humane endpoint was defined by tumor size over 1.5 cm^3^, weight loss ≥20%, or general health issues; mice were euthanized by CO_2_ and cervical dislocation.

### 2.3. Generation of Vaccine Constructs and Immunizations

The generation of the recombinant protein vaccine KISIMA-OVA and the recombinant virus VSV-GP-OVA has been described previously [25,27]. Virus titers were determined via TCID_50_ assays using BHK-21 cells. BHK-21 cells (American Type Culture Collection, Manassas, VA, USA) were cultured in Glasgow minimum essential medium (GMEM) (Gibco) supplemented with 10% FBS (PAN-Biotech), 5% tryptose phosphate broth (Gibco), 100 units/mL penicillin (Gibco), and 0.1 mg/mL streptomycin (Gibco). Mice were vaccinated with either 10^7^ or 10^8^ TCID_50_ VSV-GP-OVA (exact doses are indicated in the respective figure legends) in a prime-boost regimen with a ten-day interval. Depending on the immunization route, either 100 µL VSV-GP-OVA (intravenous (i.v.), subcutaneous (s.c.), and intraperitoneal (i.p.) inoculation) or 50 µL (intramuscular (i.m.) immunization) were applied. For vaccination with the peptide vaccine KISIMA, animals received 2 nmol KISIMA-OVA s.c. at the tail base.

### 2.4. Tissue Harvest

For the analysis of CD8^+^ tumor-infiltrating lymphocytes (TILs), tumors were harvested at the indicated time points, weighted, and not more than 1000 mg was used for further processing. Single-cell suspensions were generated by enzymatic digestion using the mouse tumor dissociation kit (Miltenyi Biotec, #130-096-730, Bergisch Gladbach, Germany) and the gentleMACS system (Miltenyi Biotec) according to the manufacturer’s specifications. Subsequently, the single-cell suspensions were filtered through a 70 µm cell strainer and erythrocyte lysis was performed using the Pharm Lyse^TM^ Lysing Buffer (BD Bioscience, #555899, San Jose, CA, USA). After a washing step with 50 mL FACS buffer (PBS containing 2% FCS, 5 mM EDTA, 0.01% sodium azide), single-cell suspensions were eluted in 100 mg/mL FACS buffer, and 200 µL was used for further analysis. For analysis of tumor-draining lymph node (tdLN) derived CD8^+^ T cells, inguinal tdLN were collected, smashed through a 40 µm cell strainer, and washed with PBS. After centrifugation, cells were eluted in 500 µL FACS buffer, of which 100 µL was used for the stainings. Tail vein blood was obtained in EDTA-coated tubes and 50 µL was used for subsequent stainings for flow cytometric analysis.

### 2.5. Flow Cytometry

Prior to the staining, samples of tumors and tumor-draining lymph nodes were incubated with Rat Serum (Jackson Immunoresearch laboratories #012-00-120, Cambridgeshire, UK), Hamster Serum (Jackson Immunoresearch, #007-00-120), and FCR block (Purified Rat Anti-Mouse CD16/CD32, BD Biosciences, #553141).

For the detection and subsequent phenotyping of antigen-specific CD8^+^ T cells in the peripheral blood, tumor tissue, and tdLNs were harvested and processed at the indicated time points as described above. Samples were incubated for 20 min at room temperature with the SINFEKL tetramer (H2-Kb SIINFEKL, MBL, #MB-TB-5001-2, Des Plaines, IL, USA) and the VSV-N tetramer (H2-Kb RGYVYQGL, MBL, #TS-M529-1). Next, the following monoclonal antibodies were used for surface staining for 30 min at 4 °C: CD90.2 (clone 30-H12), CD8 (clone 53-6.7), CD127 (clone SB/199), KLRG-1 (clone 2F1), CD19 (clone 6D5), CD14 (clone Sa14-2), CD4 (clone GK1.5), and CD45.2 (clone 104), all purchased from Biolegend (San Diego, CA, USA). Dead cells were excluded by using the LIVE/DEAD™ Fixable Near-IR Dead Cell Stain Kit (Thermo Fisher Scientific, Waltham, MA, USA). Samples of untreated mice were used to place the gates, as can be seen in Appendix A. Erythrocyte lysis was performed for blood samples as described above. To calculate the absolute number of cells, Precision Count Beads™ (Biolegend, #424902) were added shortly before acquiring the samples. Samples were measured on a FACS Canto II (BD Bioscience) and analyzed with FlowJo 10.5.3 (FlowJo LLC, Ashland, OR, USA).

### 2.6. Data Analysis and Statistics

Figure composition and statistical analyses were performed in GraphPad Prism software (GraphPad 9.0.1 Software, La Jolla, CA, USA). Statistical significance was determined by a *p*-value below 0.05, and the significant differences are indicated with asterisks in each graph (* *p* < 0.05; ** *p* ≤ 0.01; *** *p* ≤ 0.001; **** *p* ≤ 0.0001). The following statistical tests were used (as indicated in the figure legends): one-way or two-way ANOVA with Tukey’s or Sidak’s multiple comparison tests, unpaired, two-tailed *t*-test (Mann–Whitney test), and simple linear regression.

## 3. Results

### 3.1. Vaccine Dose Influences the CD8^+^ T Cell Response

Finding the optimal treatment regimen for cancer vaccines depends on many factors, e.g., the vector’s biology, the malignancy to be tackled, and possible co-therapies [7]. Viral vectors harbor the potential to activate a robust antigen-specific CD8^+^ T cell response. CTL-mediated tumor cell death occurs via additive cytotoxicity which becomes effective at a high CTL density [28], indicating that a high frequency of anti-tumoral CTLs increases treatment efficacy. However, we and others have shown that the viral vector backbones themselves induce a notable—if not dominant—immune activation [25,29,30]. This anti-viral CD8^+^ T cell response is often not considered to be important, although it may influence vaccine efficacy. To further understand the impact of the cancer vaccine vector VSV-GP on CD8^+^ T cell immunity, a comprehensive investigation was conducted in this study to decipher the factors influencing target antigen- as well as anti-viral-specific CD8^+^ T cells in terms of magnitude and quality.

Firstly, we analyzed how different doses of the vaccine vector affected the magnitude of CD8^+^ T cells. We immunized female C57BL/6JRj mice intravenously (i.v.) with either 10^7^ TCID_50_ or 10^8^ TCID_50_ of VSV-GP containing ovalbumin (OVA) as model antigen (VSV-GP-OVA). A boost immunization using the same doses was performed ten days later. CD8^+^ T cell responses were measured seven days after the prime and boost immunization in the blood (Figure 1A).

The proportions and absolute numbers of specific CD8^+^ T cells raised against the SIINFEKL epitope of ovalbumin and the dominant epitope RGYVYQGL of the VSV-derived nucleoprotein (VSV-N) were measured using tetramers for flow cytometric analysis. After the initial immunization with the lower dose of VSV-GP-OVA, around 9% and 26% of CD8^+^ T cells were elicited against OVA and VSV-N, respectively. At this time point, no significant differences were detected using the higher dose (Figure 1B, left). Interestingly, the boost immunization led to reduced VSV-N-specific CD8^+^ T cell proportions, while a statistically significant increase in OVA-specific CD8^+^ T cells was observed in the high-dose vaccination group, but not in the low-dose group. Similar trends were obtained when OVA- and VSV-N-specific CD8^+^ T cell numbers were calculated (Figure 1B, right).

These results demonstrated that VSV-GP is a potent vaccine vector for the ovalbumin model antigen and a higher antigen concentration increased the OVA-specific CD8^+^ T cell response after the boost immunization. However, it also illustrates that the viral vector elicits a strong adaptive CD8^+^ T cell immunity against itself.

### 3.2. Prime-Boost Regimen and the Vaccine Dose Influence CD8^+^ T Cell Effector Subtypes

The prime-boost regimen increased the proportion of OVA-specific CD8^+^ T cells in both frequencies and numbers. However, not only the magnitude but also the properties of CD8^+^ T cells are of relevance for immune function, and thus vaccine design [8]. Consequently, a detailed characterization of the effector subtype composition of target antigen- and virus-specific CD8^+^ T cells was performed. The gating strategy for antigen-specific CD8^+^ T cells is depicted in Figure 2A by using representative contour plots from blood seven days after the prime immunization. Corresponding controls for the used tetramers and antibodies are provided in Appendix A, respectively. The effector subtype composition was characterized among OVA-specific and VSV-N-specific CD8^+^ T cells. Here, four different phenotypes were discriminated using KLRG-1 and/or IL-7R/CD127, namely short-lived effector cells (SLECs, red), double-positive effector cells (DPs, brown), memory precursor effector cells (MPECs, green), and early effector cells (EECs, blue) [9,10].

The higher dose of VSV-GP-OVA (10^8^ TCID_50_) induced higher proportions of the SLEC subtype among OVA-specific CD8^+^ T cells after the prime and boost immunization compared to the lower dose. In contrast, the level of VSV-N-specific SLECs between groups was equal, regardless of the dose used (Figure 2B, left). Similarly, we did not detect significant differences in MPECs among VSV-N-specific CD8^+^ T cells when applying different vaccination doses, either after the priming or the boost immunization. However, seven days after the prime, MPEC proportions among OVA-specific CD8^+^ T cells were diminished in the high-dose treatment group (Figure 2B, right).

In summary, 40–60% of antigen-specific CD8^+^ T cells comprised a SLEC phenotype and only 1–6% expressed the surface markers of the MPEC subtype shortly after i.v. vaccination. Additionally, a higher dose induced a higher frequency of SLECs among OVA-specific CD8^+^ T cells.

### 3.3. Proportion of Antigen-Specific CD8^+^ T Cells Depends on the Route of Immunization

As we observed that the vaccine dose impacts the magnitude of antigen-specific CD8^+^ T cells as well as the effector phenotype, we next addressed whether different routes of immunization affect these characteristics as well. Hence, female C57BL/6JRj mice were immunized with VSV-GP-OVA (10^7^ TCID_50_) in a prime-boost regimen as depicted in Figure 3A, either intravenously (i.v.), intramuscularly (i.m.), subcutaneously (s.c.), or intraperitoneally (i.p.). Blood was collected on day 7, 17, 32, and 60 post-prime vaccination (Figure 3A). The statistical analysis for this experiment is summarized in detail in Appendix A.

After the first immunization, the i.v. and i.p. routes induced similar, significantly higher levels of antigen-specific CD8^+^ T cells compared to the i.m. and s.c. application of VSV-GP-OVA. This was observed for OVA-specific CD8^+^ T cells (Figure 3B) as well as for VSV-N-specific CD8^+^ T cells (Figure 3C). While the second application of the vaccine vector strongly increased OVA-specific CD8^+^ T cell proportions for i.v. (from ~9% to ~25%) or i.p (from ~10% to ~18%) routes, only a slight boost was measured after i.m. (from ~5% to ~8%), and there was no boost after s.c. (~5%) administration. Interestingly, the response elicited against the viral vector showed a different dynamic. Here, VSV-N-specific CD8^+^ T cell proportions decreased after the second i.v. or i.p. application, which was in line with the first experiment (Figure 1B). In contrast to the i.v. and i.p. routes, the i.m. and s.c. boost vaccination did not substantially increase OVA-specific CD8^+^ T cell frequencies but led to a substantial augmentation of VSV-N-specific CD8^+^ T cells (40%)

Over time, antigen-specific CD8^+^ T cell levels declined irrespective of the immunization route, and 32 days post-prime the disparity between groups weakened. The application route with the highest antigen-specific CD8^+^ T cell levels after the boost, which was detected after i.v. and i.m. immunization for OVA-specific and VSV-N-specific CD8^+^ T cells, respectively, also showed the highest levels at later time points. A further decrease was observed 60 days post-prime with negligible differences between the immunization routes. However, compared to untreated animals, anti-viral CD8^+^ T cells were still significantly expanded in the i.m. and OVA-specific CD8^+^ T cells in the i.v. and i.p. treatment groups (Figure 3B,C, Appendix A).

The largest proportion with 40–80% of antigen-specific CD8^+^ T cells differentiated into SLECs after the first immunization. Among OVA- and VSV-N-specific CD8^+^ T cells the i.m. and s.c. routes induced a significantly higher SLEC subtype compared to i.v. and i.p. (Figure 3D,E) post-prime. This characteristic changed after the boost immunization when the i.v. route displayed significantly higher OVA-specific SLEC levels compared to all other groups, while around 61–65% of VSV-N-specific SLECs were found in all groups, irrespective of the treatment regimen.

Even though the proportion of vector backbone-specific CD8^+^ T cells diminished (Figure 3C), VSV-N-specific CD8^+^ T cells kept a predominant SLEC phenotype over time. In contrast, OVA-specific CD8^+^ T cells lost their SLEC-specific marker KLRG-1. The dynamic of the SLEC subset was comparable between the application routes. The MPEC subset increased after the i.m., s.c., and i.p. boost immunizations, but not after the i.v. boost (Figure 3F,G). In general, MPEC levels dropped at day 32 post-prime/day 22 post-boost but were elevated again at the last analyzed time point, especially among OVA-specific CD8^+^ T cells.

Altogether, we found that not only the characteristics of the vector but also the route of immunization has the potential to shape the CD8^+^ T cell response in terms of magnitude and effector subtype composition. Furthermore, the results showed that the second immunization enhanced the memory response, boosted the antigen-specific CD8^+^ T cell response for certain immunization routes, and revealed an inverse correlation between transgene-specific and vector-specific CD8^+^ T cells seven days after the boost vaccination (Appendix A).

### 3.4. The Presence of a Target Antigen-Expressing Tumor Impacts Tumor-Antigen-Specific CD8^+^ T Cell Magnitudes as Well as Their Phenotype

As the current therapeutic development of VSV-GP as either an oncolytic or cancer vaccine virus targets existing tumors, we next examined how the presence of a tumor influences antigen-specific CD8^+^ T cells and their effector subtype composition. To minimize potentially confounding effects due to oncolytic antigen release and replication-mediated amplification of the vaccine vector, we chose the B16 mouse melanoma model, which has been shown to restrict VSV and VSV-GP replication in vivo [31,32]. As shown in Figure 4A, B16 melanoma cells expressing ovalbumin (B16-OVA) were implanted and five days later the first i.v. application of VSV-GP-OVA (10^7^ TCID_50_) was administered, which was followed by a boost ten days later. Analysis of the blood was performed seven days after each immunization. Furthermore, tumor tissue was collected 10 days post-boost. First, we compared OVA-specific CD8^+^ T cells in the blood of tumor-free, vaccinated mice (VSV-GP-OVA) and tumor-bearing animals (VSV-GP-OVA (TBA)) either untreated (Ctrl (TBA)) or VSV-GP-OVA treated (TBA) (Figure 4B, left). While no antigen-specific CD8^+^ T cells were detected in the blood of untreated mice, around 42% of OVA-specific CD8^+^ T cells were observed in the VSV-GP-OVA (TBA) group already after the prime which was significantly increased compared to the pure vaccine setting eliciting around 9% of OVA-specific CD8^+^ T cells. These populations slightly expanded after the second virus treatment, although a statistically significant increase between prime and boost immunization was observed only in the tumor-free vaccination group. The response in the tumor-free vaccine setting (VSV-GP-OVA) was significantly lower for all analyzed time points. Besides the higher frequency of OVA-specific CD8^+^ T cells in the presence of B16-OVA tumors, the SLEC effector type was also significantly higher (~74%) after the prime immunization compared to the VSV-GP-OVA group (~47%) shown in Figure 4B (middle). However, after the boost, the SLEC proportions approached comparable levels between tumor-free and tumor-bearing mice. Nevertheless, regardless of the presence of the tumor, both VSV-GP-OVA treatment groups showed a significantly increased SLEC proportion among OVA-specific CD8^+^ T cells after the boost compared to the prime immunization. Blood-derived MPEC levels did not vary between both treatment groups after the prime, but a slight reduction in OVA-specific MPEC frequencies was observed in VSV-GP-OVA (TBA) vaccinated animals after the boost immunization (Figure 4B, right).

Regarding the anti-viral CD8^+^ T cell response in blood, we observed that a higher OVA-specific CD8^+^ T cell response in VSV-GP-OVA (TBA) treated mice diminished the VSV-N-specific CD8^+^ T cell proportion significantly compared to the pure vaccine setting at all time points analyzed (Figure 4C). In contrast to OVA-specific SLECs, VSV-N-specific SLECs did not differ between tumor-bearing and non-tumor-bearing VSV-GP-OVA vaccinated animals. Similar to OVA-specific SLECs, SLECs among VSV-N-specific CD8^+^ T cells were augmented after the boost immunization regardless of the tumor presence (Figure 4C, middle). In addition, it seems that the presence of a tumor led to a slight increase in VSV-N-specific MPECs after the prime immunization (Figure 4C, right).

For successful cancer treatment, tumor-antigen-specific CD8^+^ T cells must infiltrate the tumor efficiently. Thus, we analyzed tumor-infiltrating lymphocytes (TILs) at day 10 post-boost. The tumor growth until the harvest day is shown in Appendix A. As shown in Figure 4D (left), tumors of untreated animals exhibited around 6% OVA-specific CD8^+^ TILs among CD45^+^ cells. The i.v. treatment with VSV-GP-OVA increased this proportion to around 23% (Figure 4D, left). The pre-existing OVA-specific CD8^+^ TILs among CD45^+^ TILs in untreated tumors displayed a lower proportion of SLECs compared to the newly primed antigen-specific CD8^+^ TILs after VSV-GP-OVA application (Figure 4D, middle) whereas the proportion of MPECs did not vary significantly (Figure 4D, right).

Tumors of VSV-GP-OVA vaccinated animals were populated with ~7.5% VSV-N-specific CD8^+^ TILs among CD45^+^ cells (Figure 4E, left) and ~23% OVA-specific CD8^+^ TILs among CD45^+^ cells (Figure 4D, left). Anti-tumoral OVA-specific TILs demonstrated a significantly higher proportion of SLECs compared to antiviral TILs (Appendix A). In addition, these VSV-N-specific TILs showed a significantly higher proportion of MPECs compared to OVA-specific TILs (Appendix A).

These results indicate that the presence of an antigen-expressing B16-OVA tumor may lead not only to an antigen priming effect and increased antigen-specific CD8^+^ T cells upon VSV-GP-OVA vaccination, but also may redirect the subtype composition towards the SLEC phenotype in the tumor as well as the peripheral blood. Moreover, tumor-infiltrating anti-viral bystander CD8^+^ TILs expose a distinctive CD8^+^ T cell effector subtype composition with a diminished SLEC proportion compared to tumor-restricted OVA-specific CD8^+^ T cells.

### 3.5. Heterologous Vaccine Combination Increases Antigen-Specific CD8^+^ T Cell Levels and Changes the Effector Phenotype

Cancer vaccinations face several hurdles along their path to successful clinical application. One of these hurdles is insufficient antigen-specific CD8^+^ T cell responses due to vector limitations (e.g., neutralizing antibodies or immunodominance of vector-specific CD8^+^ T cells). In addition, the inflammatory environment induced by a vaccine vector plays a major role in the CD8^+^ T cell activation and consequently effector subtype composition [17]. Thus, the heterologous combination of two different vaccine vectors might reduce some of the single-vector limitations and diversify the immune activation profile of induced CD8^+^ T cells which could ultimately translate into improved therapeutic efficacy.

We have previously shown that the combination of a peptide vaccine (KISIMA) together with VSV-GP induces a high frequency of OVA-specific CD8^+^ T cells and long-term survival in combination with an immune checkpoint inhibitor in B16-OVA tumor-bearing mice [25]. Hence, this begs the question of how this heterologous cancer vaccine regimen influences the CD8^+^ T cell composition in comparison to the homologous application of VSV-GP-OVA.

B16-OVA cells were implanted into mice and animals were primed after five days with either 2 nmol KISIMA-OVA (s.c.) or 10^7^ TCID_50_ VSV-GP-OVA (i.v). After ten days, VSV-GP-OVA was administered in both groups. Blood was analyzed seven days after the prime and boost immunization. The tumor tissue and tumor-draining lymph nodes (tdLN) were analyzed ten days post-boost (Figure 5A). The tumor growth until the harvest day is shown in Appendix A.

In peripheral blood, VSV-GP-OVA (V) prime immunization induced significantly higher OVA-specific CD8^+^ T cell proportions compared to KISIMA-OVA (K), and the heterologous combination (K-V) reached similar responses (Figure 5B, left) compared to two virus applications (V-V) post-boost. Similarly, VSV-N-specific CD8^+^ T cells did not differ significantly between treatment groups in the blood after the boost immunization (Figure 5C, left). Regarding peripheral CD8^+^ T cell effector subtypes after the prime immunization, VSV-GP-OVA induced significantly higher SLEC levels while KISIMA-OVA displayed a trend for increased MPEC levels among OVA-specific CD8^+^ T cells (Figure 5B, middle and right). After the boost immunization, no significant differences in effector phenotypes were observed between groups. Concerning the VSV-N-specific CD8^+^ T cell effector subtype composition, no alterations were detected at any analyzed time point between different treatment groups as can be seen in Figure 5C (middle and right). However, the V-V group displayed an augmented SLEC phenotype among VSV-N-specific CD8^+^ T cells, even though the proportions of VSV-N-specific CD8^+^ T cells were not increased after the boost immunization.

Even though comparable levels of peripheral OVA-specific and VSV-N-specific CD8^+^ T cell frequencies were observed in K-V and V-V treated animals, this trend did not persist at the tumor site. Here, significantly higher proportions of OVA-specific CD8^+^ T cells among CD45^+^ cells were detected in K-V compared to V-V treated mice (Figure 5D, left) while VSV-N-specific CD8^+^ TILs among CD45^+^ cells were significantly lower in the heterologous compared to the homologous treatment group (Figure 5E, left). Intriguingly, VSV-N-specific CD8^+^ TIL effector subtypes were equivalent between treatment groups (Figure 5E, middle and right), whereas OVA-specific intratumoral SLECs were slightly diminished in K-V compared to V-V treated animals, as can be seen in Figure 5C (middle). Although not statistically significant, a trend for a marginally higher MPEC population among OVA-specific CD8^+^ TILs in the K-V group was observed (Figure 5B, right). In addition, similar to V-V-derived intratumoral phenotypes, (Appendix A), K-V-derived intratumoral SLECs differed between antigen specificities. As shown in Appendix A, OVA-specific TILs displayed significantly higher SLEC proportions compared to VSV-N-specific TILs. In contrast to V-V, K-V-derived MPEC proportions did not vary between OVA- and VSV-N-specific TILs.

Similar to the findings among TILs, the highest proportion of tdLN-derived OVA-specific CD8^+^ T cells was observed in K-V compared to V-V treated mice (Figure 5F, left). In contrast, virus-specific CD8^+^ T cells were similarly increased in both treatment groups compared to controls (Figure 5G, left). Regarding intranodal OVA- and VSV-N-specific CD8^+^ T cell effector subtypes, no statistically significant differences were detected between treatment groups for SLEC (Figure 5F,G, middle) or for MPEC (Figure 5F,G, right) subtypes. However, it should be noted that OVA- and VSV-N-specific MPEC proportions were generally higher in tdLNs compared to peripheral blood or tumors.

Taken together, the heterologous combination of a peptide and viral cancer vaccine increased tdLN-derived and intratumoral OVA-specific CD8^+^ T cell frequencies while VSV-N-specific TILs were reduced compared to the homologous V-V regimen. In addition, intratumoral OVA-specific CD8^+^ T cells exhibited a decreased SLEC phenotype compared to the homologous virus treatment.

## 4. Discussion

Inducing a high proportion of antigen-specific CD8^+^ T cells has long been the focus of vaccine vector design. In recent years, however, considerations about qualitative in addition to quantitative changes in CD8^+^ T cells have gained more attention. CD8^+^ T cells exert their effector functions through a variety of mechanisms which include the direct killing of target cells, self-amplification capacities, and the capability to coordinate immune responses via the secretion of chemo-attractants [8]. Characteristics defining CD8^+^ T cell quality comprise polyfunctionality, exhaustion status, and CD8^+^ T cell phenotypes. One way to classify phenotypes is to distinguish CD8^+^ T cells into short-lived effector cells (SLECs) or memory precursor effector cells (MPECs) via the surface expression of KLRG-1 or IL-7R, respectively [9]. A high number of SLECs is desirable shortly after a pathogen encounter to rapidly control the infection and diminish the pathogen load. In contrast, high levels of MPECs enable better memory formation and thus long-term protection. However, whether one or the other phenotype can be associated with better disease control is an unsolved question, which might vary between malignancies; opposing results have been published in this regard [16,18]. Also, it has been postulated that a heterogeneous mixture of antigen-specific CD8^+^ T cells might be ideal, although the exact ratio of effector to memory cells remains to be elucidated [33]. Thus, a detailed, basic phenotype characterization gives further insights regarding the vaccine vector’s properties, and in the future might help in choosing treatment regimens suitable for distinctive diseases.

We demonstrated in our study that an increased vaccine dose elevated the overall OVA-specific CD8^+^ T cell frequency as well as the SLEC percentage among OVA-specific CD8^+^ T cells after the prime as well as the boost immunization, while the lower dose displayed an increased MPEC frequency after the prime immunization. This is in line with prior studies that showed an inverse correlation of MPEC proportion with antigen-specific CD8^+^ T cell frequency [15]. Furthermore, an increase in SLEC frequency was observed with elevated nanoparticle vaccine dose. However, this boosting effect reached a plateau after certain concentrations where neither CD8^+^ T cell frequencies nor SLEC proportions were further augmented. In addition, in separate studies the highest Ebola GP-specific CD8^+^ T cell response was measured after vaccination with the highest dose of VSV-EBOV [34]. Interestingly, many studies have tried to decipher the impact of vaccination dose on CD8^+^ T cell activation dynamics and memory formation. For example, a lower antigen dose has been associated with better memory formation, whereas vaccination with a high peptide dose seems to favor early and strong in vivo expansion and differentiation of tumor antigen-specific CD8^+^ T cells [35].

Understanding the optimal immunization route for a robust immune response is paramount for successful vaccinations against pathogens or cancer. Conflicting results have been published dealing with the issue that immunization routes alter the innate and adaptive immune system in different ways which might be unique for individual vectors. Of note, different routes of vaccination may present with variations in the T cell kinetic, and selecting a single time point for analysis may introduce a certain bias towards one group over the other, which may be an experimental limitation of many of the comparative studies. While some studies with nanoparticle vaccines reported that the innate or adaptive immune responses do not differ between i.m. and s.c. applications [36], we found that i.v. prime-boost vaccination with VSV-GP-OVA generated the highest OVA-specific CD8^+^ T cell response together with a reduced anti-viral response compared to i.p., i.m., and s.c. immunization. These findings confirm previously published results that i.v. immunization with a peptide vaccine was superior compared to the i.m. and s.c. route in terms of tumor-specific CD8^+^ T cell frequencies and therapeutic efficacy [37]. In contrast, others have shown that for a nanoparticle vaccine, s.c. immunization is superior in terms of antigen-specific CD8^+^ T cell frequencies compared to the i.v. immunization route [15]. These varying results highlight the importance of deciphering the optimal treatment route for each vaccine vector individually. Another noteworthy observation is the significantly higher proportion of OVA-specific CD8^+^ T cells together with a lower proportion of VSV-N-specific CD8^+^ T cells which was induced in the i.v. and i.p. immunization route compared to the s.c. and i.m immunization route. One possible hypothesis for this increased transgene-specific response is that it could originate from the biological properties of the virus. While the nucleoprotein (VSV-N) is incorporated into VSV-GP’s virion at high amounts (approx. 1200 per virion), the foreign transgene ovalbumin is exclusively encoded in the viral negative-strand RNA genome and its production starts upon host cell infection [38]. Thus, productive viral replication may presumably increase the antigen amount of ovalbumin, thereby enhancing successful antigen presentation and the priming of naïve OVA-specific CD8^+^ T cells.

Specific CD169^+^ marginal zone macrophages (MZM) in the spleen and lymph nodes allow so-called enforced viral replication of VSV, LCMV, and other lymph-borne viruses via USP18 expression [39,40]. Furthermore, these macrophages can transfer antigen to cDC1s, which cross-present it to and activate primary and memory CD8^+^ T cells [41,42]. Recently, these findings were also corroborated for the genetically engineered chimeric VSV-GP despite its altered cellular tropism compared to parental VSV [43]. In this study, VSV-GP also infects, replicates, and lyses CD169^+^ MZM in the spleen, mandibular, and inguinal lymph nodes. Furthermore, it was demonstrated that intravenous injection of high-dose VSV-GP reaches the spleen of mice without much delay. In the spleen, the transcription of positive-strand viral mRNAs starts as early as one hour post-infection and leads to transient viral replication within the first 24 h. This active replication could explain the higher OVA-specific CD8^+^ T cell responses following i.v. immunization, while the underlying mechanism for the i.p. route is not that clear. After i.p. immunization, molecules up to a size of 5000 molecular weight are proposed to enter the circulation rapidly [44], which may also apply to VSV-GP. Before the particles enter the systemic circulation and access the spleen, they pass through a complex network of visceral lymph nodes. This circulation would provide the chance to capture viruses in CD169^+^ subcapsular sinus (SCS) macrophages, thereby allowing enforced viral replication [40,43]. Taken together, these findings highlight the importance of addressing unique vaccine vector characteristics and their ability to influence immunogenicity when developing vaccine regimens for various diseases.

Concerning the variations of effector phenotypes between immunization routes, limited published data is available for comparison. Nevertheless, some findings confirm our observations for the i.v. immunization route. For example, a study using VSV-OVA reported the induction of similar OVA-specific CD8^+^ T cell proportions after i.v. vaccination in the blood [17]. The higher SLEC frequencies that we saw could be explained by the higher immunization dose used in our experiments. For VSV-N-specific CD8^+^ T cell effector subtypes after i.v. infection, the published data are in line with our findings, showing around ~45% SLEC and <10% MPEC seven days after the prime immunization [45], although the vaccination doses again differed between experiments. Of note, VSV-GP shares the nucleoprotein with its parental VSV strain. Hence, antiviral N epitopes are expected to be comparable. Remarkably, varying vaccination doses between our findings and these published results led to differences in OVA-specific CD8^+^ T cell effector subtypes, but not VSV-N-specific effector subtypes. Intriguingly, this matches with our observation that the vaccine dose alters the OVA-specific but not VSV-N-specific effector subtype composition (Figure 2B) following systemic immunization. One hypothesis for this discrepancy comprises the observations of others that CD8^+^ T cell magnitudes and SLEC phenotypes increase steadily with increasing antigen availability until a plateau is reached where no further augmentation of immune response can be observed [15,46]. In our case, VSV transcription follows a sequential gradient, in which genes located closer to the 3′ end of the genome are produced in higher amounts than their subsequent neighbors [47]. In our construct, VSV-N is located at position 1, whereas the foreign transgene ovalbumin is inserted in the second-last position number 5. Thus, we could speculate that the threshold for maximum CD8^+^ T cell activation is reached at lower virus doses for VSV-N in comparison to OVA. Another possible explanation for the increased VSV-N response after the prime immunization could be a delayed antigen presentation of the OVA peptide due to the later expression in infected cells compared to the VSV-N protein that is part of the viral vector.

However, the exact underlying mechanism for this discrepancy between antigens remains a matter for future investigations. In summary, we here present a comprehensive characterization of a wide range of immunization routes in a homologous prime-boost vaccine regimen concerning not only antigen-specific CD8^+^ T cell frequencies but also qualitative markers like effector subtypes.

Additionally, our study provides insights into whether and how the pre-existence of a target antigen source in the host (such as an OVA-expressing tumor) influences CD8^+^ T cell quantity and quality. In detail, we observed that the presence of B16-OVA tumors may serve as a sort of pre-prime, as we detected intratumoral OVA-specific CD8^+^ T cells in untreated animals and this measurable OVA-specific T cell population was boosted upon first VSV-GP-OVA vaccination. Consequently, OVA-specific CD8^+^ T cell frequencies and the SLEC proportion among these cells were increased following VSV-GP-OVA prime immunization compared to vaccinated non-tumor-bearing animals. Furthermore, in this B16 tumor model, we suppose that immunogenicity changes reflect predominantly VSV-GP’s vaccine vector properties rather than being affected by a potential lytic and replicative activity of the vector in these tumors. B16 has been well characterized to strongly limit the in vivo activity of VSV or VSV-GP variants, in part due to its potent antiviral type-I interferon response [25,31,32,48]. Regarding the anti-viral immune response between tumor-bearing and tumor-free animals, we observed diminished VSV-N responses in tumor-bearing animals. Mechanistically, the presence of the tumor induced a fraction of endogenously activated, OVA-specific CD8^+^ T cells, which may have led to an advantage in epitope competition over the smaller VSV-N-specific CD8^+^ T cell precursor pool [49]. It is known that the precursor frequency is a critical aspect of the CD8^+^ T cell response to infections and that the pre-existing response to one epitope within a polyepitope construct can significantly impair the ability of a vaccine to expand cytotoxic T cells of the other specificities [50,51]. These findings corroborate published results from other tumor models. We have previously shown that the presence of the colorectal MC-38 tumor impacts the frequency and dominance pattern of a multi-epitope neoantigen vaccine [26]. Moreover, in the present study, we found that the presence of a tumor also impacts the phenotype of antigen-specific CD8^+^ T cells by increasing the SLEC proportion among these cells after the prime immunization.

Another compelling observation was the difference in CD8^+^ T cell phenotypes among intratumoral OVA- and VSV-N-specific CD8^+^ T cells with the latter displaying diminished SLEC frequencies together with increased MPEC frequencies. Due to the negligible contribution of oncolysis and intratumoral virus propagation in the B16-OVA model, we presume that the anti-viral CD8^+^ T cells are largely nonspecific bystander TILs. This premise would explain their reduced SLEC phenotype due to the absence of local antigen recognition and subsequent TCR signaling and activation. In the context of cancer, non-specific bystander recruitment occurs as a consequence of the inflammatory environment and local IFNɣ-producing CD8^+^ T cells [52,53]. It has been reported recently that virus-specific bystander TILs derived from a chronic infection often exhibit a non-exhausted memory phenotype and consist predominantly of memory and effector memory CD8^+^ T cells, whereas tumor-restricted TILs display mostly an acutely activated, effector memory, precursor-exhausted, or a terminally exhausted CD8^+^ T cell phenotype [54]. These findings are in line with the diminished SLEC and increased MPEC frequencies that we observed for anti-viral TILs. Interestingly, in our study, anti-viral bystander TILs were elicited by a recent infection (10 days before tumor harvest) in contrast to the above-mentioned chronic infection-based Epstein–Barr virus (EBV)-specific bystander TILs. However, we detected these phenotypic changes in anti-viral TILs in comparison to tumor-restricted OVA-specific TILs.

After heterologous vaccination with a peptide vaccine and VSV-GP-OVA in the B16-OVA model, we saw an increased transgene-specific response in the tumor and tdLN as we have shown previously in other tumor models [25,26]. A high systemic frequency of OVA-specific CD8^+^ T cells was observed for the homologous as well as the heterologous treatment regimen. However, only the heterologous treatment combination was able to direct these immune cells in high frequencies to their site of action in the tumor tissue. Whether different CD8^+^ T cell effector phenotypes possess the capability to influence treatment efficacy is beyond the scope of this study. In this regard, others have recently postulated that the heterogeneous phenotype mixture of antigen-specific CD8^+^ T cells might be a better predictor for the treatment efficacy of cancer vaccines, but that the exact composition might differ between tumor settings [8,33]. Furthermore, we are aware that immunological studies using the ovalbumin antigen have limited comparability to the clinical setting. However, especially in homologous prime-boost treatment regimens, this antigen is the best option to induce an antigen-specific CD8^+^ T cell response sufficient enough to perform statistically reliable phenotype characterizations.

We here provide a basic description of CD8^+^ T cell effector phenotypes in a pre-clinical model. Whether this phenotype classification can be transferred to the human setting of the currently ongoing clinical trials remains a matter of debate and experimental explorations. Nevertheless, others have shown that these surface markers can indeed describe functionally distinct CD8^+^ T cell populations in humans after chronic infections (cytomegalovirus (CMV) and EBV) and acute influenza infection [55]. Furthermore, it has been recently observed that the presence of chronic lymphocytic leukemia skews CD8^+^ T cells towards the SLEC while diminishing the MPEC phenotype [56].

In conclusion, this work examined how multiple factors such as dose, route, presence of an antigen-expressing tumor entity, and heterologous vaccine partners influence the frequency and effector subtype composition of transgene- and vector-specific CD8^+^ T cells in mice, thereby providing a basic qualitative CD8^+^ T cell characterization of VSV-GP.

Nevertheless, it is equally important to obtain a better understanding of underlying immunological mechanisms, especially for challenging pathogens, chronic infections, and cancer vaccines.

Consequently, a better understanding of the quality, which includes T cell phenotypes of vaccine-induced immune cells, provides one strategy to ameliorate current vaccine platforms and heterologous vaccine partners.

## Figures and Tables

**Figure 1 vaccines-12-00867-f001:**
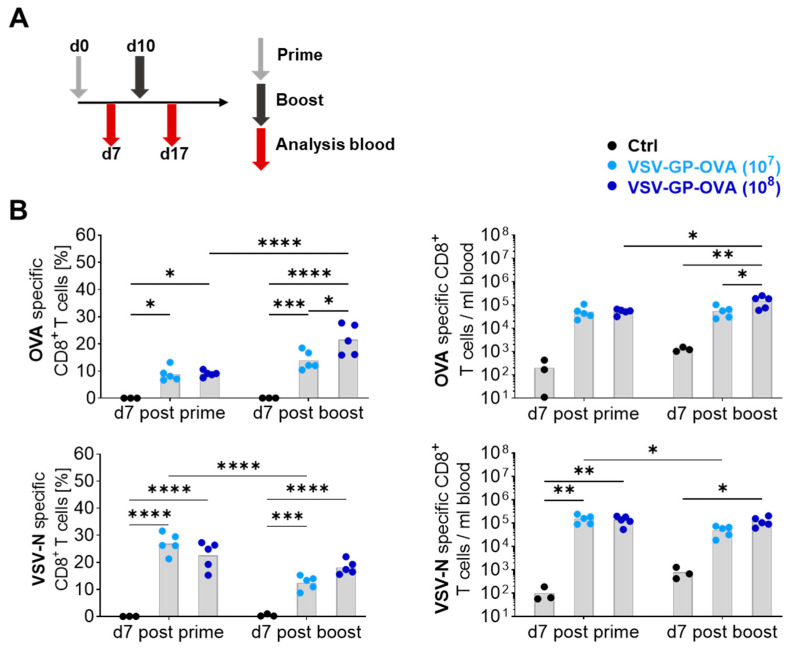
Immunization with a higher dose of VSV-GP-OVA increases the OVA-specific CD8^+^ T cell response, without increasing the VSV-N-specific response post-boost. (**A**) Schematic schedule of the experimental setup. Six- to eight-week-old female C57BL/6JRj mice were immunized with 10^7^ or 10^8^ TCID_50_ VSV-GP-OVA in a 10-day prime-boost regimen. Seven days after each immunization, blood was collected for flow cytometry analysis. (**B**) Proportions (left) and absolute numbers (right) of blood-derived OVA- (upper graphs) and VSV-N-specific (lower graphs) CD8^+^ T cells are depicted. Grey columns indicate mean values. Two-way ANOVA followed by Tukey’s multiple comparison test was performed. Significant differences between the groups (*n* = 3 for Control, *n* = 5 for treatment groups) are indicated with asterisks (* *p* ≤ 0.05, ** *p* ≤ 0.01, *** *p* ≤ 0.001, **** *p* ≤ 0.0001). This experiment was performed once.

**Figure 2 vaccines-12-00867-f002:**
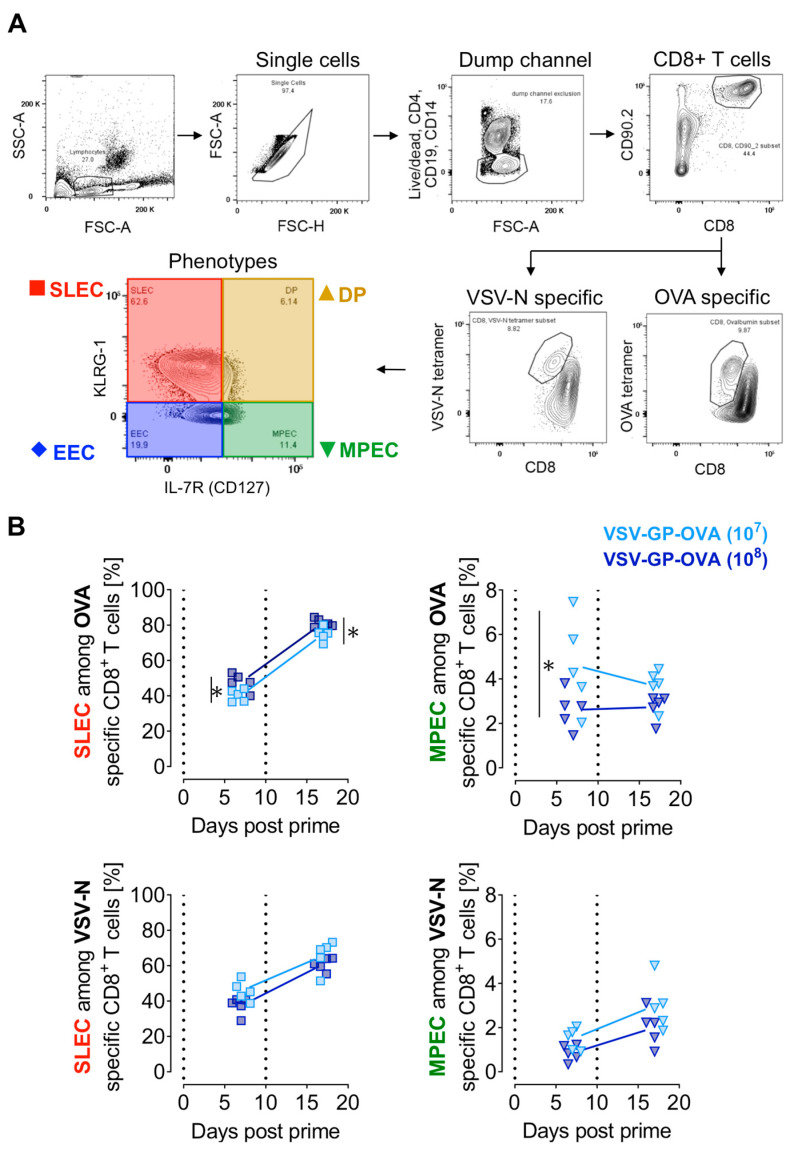
Immunization with a higher dose of VSV-GP-OVA slightly increases SLEC frequency among OVA-specific CD8+ T cells. (**A**) Gating strategy for the detection of antigen-specific CD8^+^ T cell effector subtypes in peripheral blood. CD8^+^ T cell effector subtypes are divided into CD127^-^KLRG1^-^ early effector cells (EECs, highlighted in blue rhombus), CD127^-^KLRG1^+^ short-lived effector cells (SLECs, red square), CD127^+^KLRG1^-^ memory-precursor effector cells (MPECs, green down triangle), and CD127^+^KLRG1^+^ double-positive cells (DPs, brown up triangle). (**B**) Mice were immunized with either 10^7^ or 10^8^ TCID_50_ VSV-GP-OVA two times in a 10-day interval as shown in Figure 1A. The fractions of SLECs and MPECs among OVA- and VSV-N-specific CD8^+^ T cells on day 7 post-prime and day 7 post-boost are shown. The dotted lines indicate the time of vaccination. Two-way ANOVA followed by Šidák multiple comparison test was performed. Significant differences between groups (*n* = 5) are indicated with asterisks (* *p* ≤ 0.05). This experiment was performed once.

**Figure 3 vaccines-12-00867-f003:**
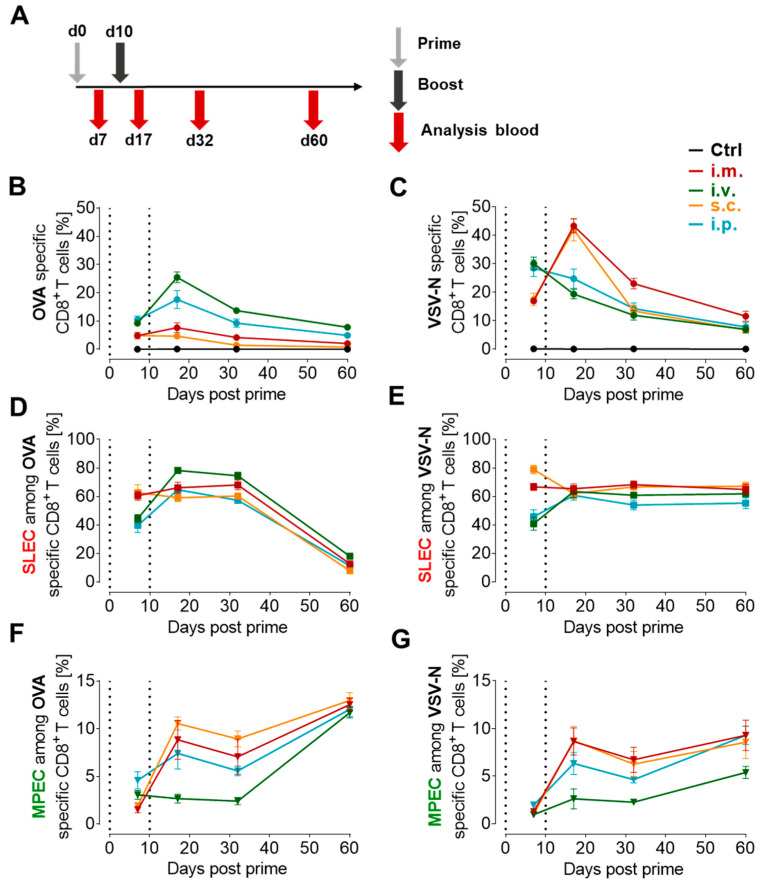
Different immunization routes modulate the ratio of OVA-specific and VSV-N-specific CD8^+^ T cell frequencies and affect the effector subtype composition. (**A**) Schematic schedule of the experimental setup. Six- to eight-week-old female C57BL/6JRj mice were immunized with 10^7^ TCID_50_ VSV-GP-OVA intravenously (i.v.), subcutaneously (s.c.), intraperitoneally (i.p) or intramuscularly (i.m.), followed by an equal boost immunization ten days later. On days 7, 17, 32, and 60 post-prime, blood was drawn for flow cytometry analysis. The proportion of OVA- (**B**) and VSV-N-specific (**C**) CD8^+^ T cells are shown over time. (**D**,**E**) display the proportion of short-lived effector cells (SLECs, KLRG-1^+^CD127^-^) among OVA and VSV-N-specific CD8^+^ T cells, respectively. (**F**,**G**) depict memory-precursor effector cells (MPECs, CD127^+^KLRG-1^-^) among OVA- and VSV-N-specific CD8^+^ T cells. The dotted lines indicate the time of vaccination. The experiment was performed once (*n* = 5 per group) and data is shown as mean ± SEM. Details for statistical analysis can be found in Appendix A.

**Figure 4 vaccines-12-00867-f004:**
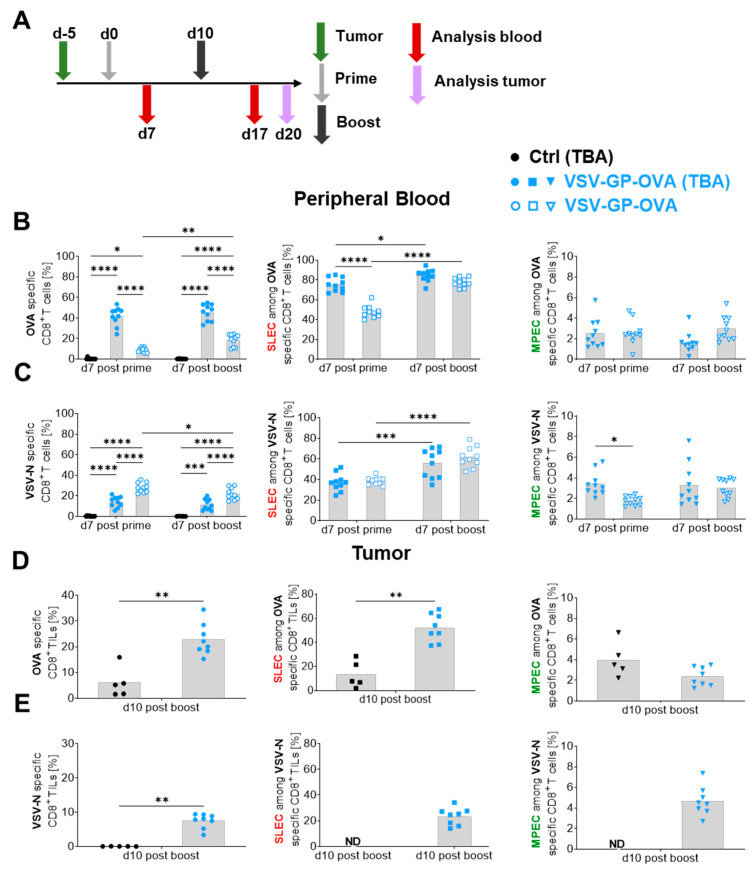
The presence of a B16-OVA tumor increases OVA-specific CD8^+^ T cell frequencies, changes effector subtype composition after vaccination, and intratumoral OVA- and VSV-N-specific CD8^+^ T cells exhibit different effector phenotypes. (**A**) Schematic schedule of the experimental setup. B16-OVA tumor cells were subcutaneously (s.c.) implanted into six- to eight-week-old female C57BL/6JRj mice (TBA, tumor-bearing animal). Mice were either untreated (Control (Ctrl) TBA) or treated five days later in a 10-day prime-boost interval with 10^7^ TCID_50_ VSV-GP-OVA (VSV-GP-OVA (TBA)). Additionally, tumor-free mice received the same prime-boost immunizations (VSV-GP-OVA). Seven days after each immunization blood was collected for flow cytometry analysis, and twenty days post-prime tumors were isolated for tumor-infiltrating lymphocyte (TILs) analysis. The fraction of blood-derived OVA- (**B**) or VSV-N-specific (**C**) CD8^+^ T cells and the percentage of short-lived effector cells (SLECs) as well as the memory-precursor effector cells (MPECs) among those cells are illustrated. The proportion of intratumoral OVA- (**D**) and VSV-N-specific (**E**) CD8^+^ T cells among CD45^+^ cells as well as the percentage of SLECs (middle, squares) and MPECs (right, down triangles) of each antigen-specific subset are shown. Grey bars depict mean values. Two-way ANOVA followed by Tukey’s multiple comparison test was performed for the analysis of (**B**,**C**) with *n* = 10 per group. An unpaired, two-tailed *t*-test (Mann–Whitney) was performed in (**D**,**E**) with *n* = 6 for Control and *n* = 8 for VSV-GP-OVA (TBA). (**A**–**E**) show results from two independent experiments and significant differences between the groups and time points are indicated with asterisks (* *p* ≤ 0.05, ** *p* ≤ 0.01, *** *p* ≤ 0.001, **** *p* ≤ 0.0001). ND = not determined. Tumor growth is shown in Appendix A.

**Figure 5 vaccines-12-00867-f005:**
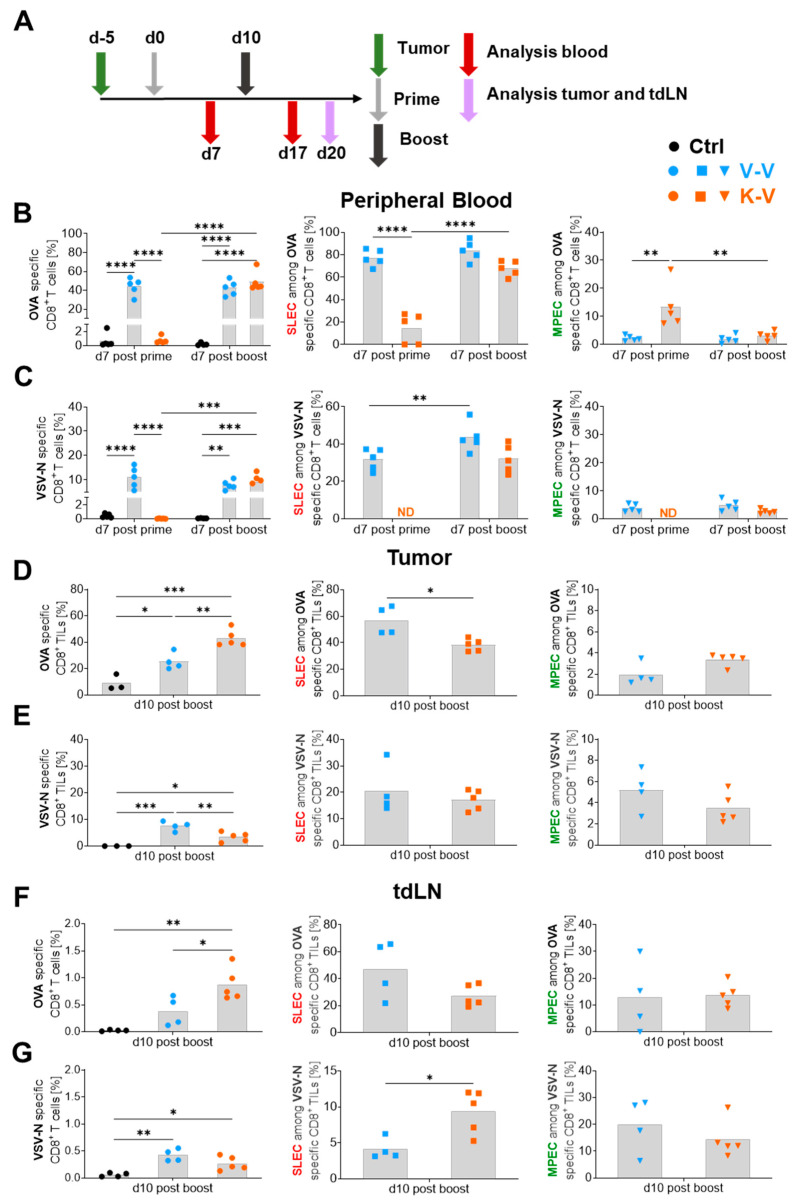
Heterologous vaccination increases OVA-specific CD8^+^ T cells in the tumor and tumor-draining lymph nodes and influences CD8^+^ T cell effector subtype composition (**A**) Schematic schedule of the experimental setup. B16-OVA tumor cells were subcutaneously (s.c.) injected into six- to eight-week-old female C57BL/6JRj mice. Mice were treated five days later in a ten-day prime-boost interval with 10^7^ TCID_50_ VSV-GP-OVA (V) and 2 nmol KISIMA-OVA (K) in either a homologous (V-V) or heterologous (K-V) combination. Seven days post-treatment, peripheral blood was collected, and ten days post-boost tumors and tumor-draining lymph nodes (tdLN) were harvested for flow cytometric analysis. The percentage of OVA-specific or VSV-N-specific CD8^+^ T cells and the SLEC (squares) and MPEC (down triangles) proportions among those cells from peripheral blood are shown in (**B**,**C**), in the tumor among CD45^+^ cells in (**D**,**E**) and in the tdLN in (**F**,**G**), respectively. Grey bars depict mean values. Two-way ANOVA followed by Tukey’s multiple comparison test was performed in (**B**,**C**). One-way ANOVA followed by Tukey’s multiple comparison test was performed in the left graphs of (**D**–**G**) and an unpaired, two-tailed *t*-test (Mann–Whitney) was performed for the middle and right graphs. The experiment was performed once with *n* = 5 per group in (**B**,**C**) and *n* = 3 in Control (Ctrl), *n* = 4 in V-V, and *n* = 5 in KV for (**D**–**G**). Tumor growth is shown in Appendix A. Part of this dataset has already been shown in Figure 4 and was re-analyzed for statistical comparison to discuss another scientific question. Significant differences between the groups and time points are indicated with asterisks (* *p* ≤ 0.05, ** *p* ≤ 0.01, *** *p* ≤ 0.001, **** *p* ≤ 0.0001). ND = not determined. Tumor growth is shown in Appendix A.

## Data Availability

Pertinent data to support the findings of this study are included in the manuscript and Appendix A. Further data supporting the findings are available upon request from the corresponding author.

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
