# Peer review of "Characterization of the Anti-Viral and Vaccine-Specific CD8+ T Cell Composition upon Treatment with the Cancer Vaccine VSV-GP"

_vaccines, 2024, doi:10.3390/vaccines12080867_

Round 1

Reviewer 1 Report

Comments and Suggestions for Authors

The article "Characterization of the anti-viral and vaccine-specific CD8+ T cell composition upon treatment with cancer vaccine VSV-GP" presented by T. Hofer et al. describes valuable experimental data on the characteristics of CD8+ T cell subpopulations elicited following vaccination using homologous VSV vector prime/boost or combined peptide/VSV prime/boost regimen.

Although the article is highly informative, well-written and contains valuable experimental data, I would recommend its revision before publication due to the following considerations:

Major issues:

1. While in the Materials and Methods section it is clearly stated that the tumor-bearing mice were monitored throughout the experiment and tumor volume was assessed (with the formula provided and 1.5 cm^3 tumor volume mentioned as humane endpoint), these data are not presented in the manuscript. The data on the differences in tumor growth kinetics between the study groups would be highly valuable for understanding of the functional activity of the described cell populations and thus should be included in the article.

2. It is legitimately stated in the Introduction section that upon booster vaccination the peak clonal expansion of T cells occurs faster than upon prime. Hence, it is questionable whether direct comparison should be performed between the proportions of antigen-specific T cells and the CD8+ T cell subpopulation composition between day 7 post prime and day 7 post boost. Moreover, the bias of the time point used for the T cell response assessment is also applicable to Figure 3, where the comparison between four immunization routes is performed. The time points of peak specific CD8+ T cell levels are highly likely to be different for the four vaccine administration routes described, therefore their uniform assessment at day 7 post prime might be misleading. This bias is supposedly less pronounced at boost, as the selected clones are already present. I believe that this consideration does not reduce the overall value of the presented findings, but it should be mentioned as a limitation of the study design.

2.1. It is stated in the Discussion section that while the VSV-N is a structural protein and is thus interacting with APCs from the moment of vaccine injection, the ovalbumin is encoded in the RNA. Besides the amount of the antigen produced, this difference presumably also shifts the antigen presentation of OVA to a later time point compared to the VSV-N. This is also a limitation of the design, as it would introduce additional bias to the comparison between the CD8+ T cell response to these two antigens after prime. 

3. It is stated in the section 3.2. of the Results that "The higher dose of VSV-GP-OVA (108 TCID50) induced higher frequencies of the SLEC subtype among OVA-specific CD8+ T cells after the prime and boost immunization compared to the lower dose", but in the Figure 2 B it is clearly the opposite. The lower dose induced higher proportions of both SLEC and MPEC.

Minor remarks in the text:

1. lines 18, 25, 52, 93, 226 etc. Is the term "cell frequencies" the optimal one? I believe that "CD8+ T cell proportions" would be more appropriate.

2. line 132. Please include the manufacturer of DMEM.

3. line 140. The term "humane endpoint" would be more appropriate here instead of "end criteria".

4. line 169-188. The description is incomplete. Please specify how antigen-specific cells were defined (reference to Figure S1).

5. line 213. "quantity" instead of "magnitude"?

6. line 318. "10^7"

7. line 455. "2 nmol"

Remark for the graphs

The legends are hard to read due to the different color schemes used. Filled and empty circles, squares and triangles in Figure 4 vs triple shapes in the legend in Figure 5. Please unify

Reviewer 2 Report

Comments and Suggestions for Authors

Hofer et al. used VSV-GP to perform basic immune characterization of various factors that affect antiviral and vaccine target-specific CD8 T-cells and their phenotype. The authors tested the effect of vaccination dose, route of immunization, presence of a target antigen-expression tumor and heterologous prime-boost with a peptide vaccine (KISIMA) partner following vaccination with antigen-armed VSV-GP – all in mice. They found increasing the vaccine dose, increased the antigen specific CD8 T cell frequencies and altered the phenotype of these cells. Intravenous inoculation (followed by intraperitoneal, intramuscular, and subcutaneous route) produced more antigen specific CD8 T cells with the lowest antiviral response. Tumor specific CD8 T cells had different phenotypes than bystander CD8 T cells in the mice. Finally, the combination of peptide and viral vaccine produced the highest frequency of antigen specific CD8 T cells in the tumor and tumor-draining lymph nodes. The manuscript was very well written with appropriate background, references, and discussion provided. 

1.     Materials and Methods: please describe the BHK-21 cell culture conditions. 

Round 2

Reviewer 1 Report

Comments and Suggestions for Authors

All questions from the previous round of review have been adequately addressed by the authors. 

The significant observation in the newly presented Figure S4 is that the differences in the volume of the tumors at later time points are unexpectedly high considering inbred mice and clonal implanted cells (especially in the control group). Also, in some animals the tumor size starts to rapidly decline around day 21, which might be explained by immune rejection and potentially impact the proportions and the phenotype of T cell subpopulations within study groups described in Figures 4 and 5.

Still, as the observations mentioned above are present both in the control and the treated groups, this does not reduce the overall merit of the presented manuscript